# Using Landfill Sites and Marginal Lands for Socio-Economically Sustainable Biomass Production through Cultivation of Non-Food Energy Crops: An Analysis Focused on South Asia and Europe

Tavseef Mairaj Shah [1,*], Anzar Hussain Khan [1], Cherisa Nicholls [1], Ihsanullah Sohoo [2] and Ralf Otterpohl [1]

[1] Rural Revival and Restoration Engineering (RUVIVAL), Institute of Wastewater Management and Water Protection, Hamburg University of Technology, Eissendorfer Strasse 42, 21073 Hamburg, Germany
[2] Department of Energy & Environment Engineering, Dawood University of Engineering, Karachi 74800, Pakistan
* Correspondence: tavseef.mairaj.shah@tuhh.de

**Abstract:** Food security and energy transition are among the current major global environmental challenges. Although these issues individually are significant in their own right, they are connected to each other in a nexus with different interrelationships and dependencies. In the quest for non-fossil alternatives for energy, cultivation of bioenergy crops has become an important part of the energy policy in many countries. In this regard, the use of fertile agricultural land for growing crops for energy production rather than for food supply affects the global food security. Recent conflicts and the geopolitical crisis in Europe, leading to increased food, fuel, and fertiliser prices, the existing climate crisis, and the crisis caused due to the COVID-19 pandemic, have further reinforced the understanding of this nexus, with certain countries mulling limiting biofuel production from agricultural land and others banning food grain exports to safeguard food supply. The idea of growing non-food energy crops on marginal lands in general and closed landfill sites in particular is hence ever more relevant, to avoid land-use concurrence between food needs and energy needs. Landfilling has been the dominant waste management strategy until recently in European countries and is still the dominant mode of waste management in low-income regions like South Asia. This paper provides a review of the economic as well as environmental benefits of growing *Ricinus communis* L., *Jatropha curcas* L., and *Populus deltoides* as energy crops on closed landfill sites in the South Asian context. While as the cultivation of *Miscanthus X Giganteus*, *Silphium perfoliatum* L., and *Panicum virgatum* (Switchgrass) is reviewed in the European context. The cultivation of non-food energy crops like these on closed landfill sites and marginal lands is presented as a potential component of an integrated food-energy policy, with an increased relevance in the current times. In the current times of multiple crises, this measure is of increasing relevance as a part of the overall strategy to achieve resilience and environmental sustainability.

**Keywords:** clean energy; ecosystem services; energy policy; non-food energy crops; soil regeneration; water-energy-agriculture nexus

## 1. Introduction

The current global challenges of food and energy security are linked to each other in different ways. The FAO's yearly report on food security reports that up to 828 million people worldwide faced food insecurity in 2021, a number 20% higher than the pre-pandemic value, while as almost 3.1 billion people worldwide could not afford a healthy diet in 2020 [1]. In another timely report on land and water resource use, the FAO brings into focus the water-energy-agriculture nexus by highlighting the interconnected systems of land, soil, and water and terming energy production as a 'related system' [2]. Although

the report stops short of proposing a change in land allocation for bioenergy production, it calls for 'complementary efforts' in the energy production sector [2]. Bioenergy contributes 13.4% of the world primary energy consumption and is mainly used in parts of Asia and Africa in the form of wood and dung in rural areas [3]. The use of bioenergy involves the emission of carbon that was captured by plants during their growth cycle, which is in contrast to the use of fossil energy which leads to the emission of carbon that was stored in the earth for millions of years [3]. Today, the contribution of energy crops for bioenergy production is relatively small as compared to the other sources of energy, whereas the potential of producing energy from energy crops is estimated to reach ~400 EJ/year by 2050. In this regard, the use of productive agricultural land for bioenergy generation is a point of debate given the fragile food supply chains and food insecurity in different parts of the world. Geopolitical conflicts further bring to light the need to address these challenges from a systems perspective of the food-water-energy nexus. Current debates around energy security hence focus on using carbon-neutral, clean and environment-friendly sources of energy [3,4].

In the European context, recent geopolitical events have put into spotlight the fragility of food supply chains and their vulnerability to multiple crises like climate change, COVID-19, rising food costs and conflict [5]. The resulting shortages of food grains and edible oils led to substantial debate given significant portion of these essential commodities is used for bioenergy production. This includes 18% of world's vegetable oils used in biodiesel production and 10% of world's grains used in bioethanol production [6,7]. In the European Union (EU), these numbers are even higher with 58% of rapeseed oil, 50% of palm-oil, and 32% of soy oil being used as motor fuel, and the land used to grow biofuels a huge 14 million hectares [6,7]. Studies have reported that the crops used for biofuel production could feed billions of people worldwide, effectively ending world hunger [7–9]. In this context, it was an expected consequence that Germany mulled downsizing the cultivation of bioenergy crops on fertile agricultural land to make room for growing crops for food [10,11]. On the other hand, some countries in South Asia restricted the export of food grains with the same reasoning [12,13]. Furthermore, different issues of land degradation including soil salinization and loss of soil nutrients, pollution of surface and ground water make the choice of crops grown in agricultural land even more significant [14]. This makes the idea of using marginal lands for growing crops for energy production more pertinent in the European context, to free up at least some portion of the land area to grow crops for food supply [15,16]. With increasing focus being put on land-based climate change mitigation through decreasing greenhouse gas emissions and increasing carbon sequestration in different EU policies, cultivating non-food energy crops on marginal lands that contribute to these goals support this idea [17].

The increasing industrialisation in South Asian countries is continuously influencing the energy use patterns with increased reliance of economic growth on fossil fuels [18]. The use of energy crops for biofuel and bioenergy production can be an important economical alternative to the use of non-renewable sources of energy in such a scenario [4]. However, extensive use of agricultural land for energy crop cultivation might result in different environmental issues in addition to sparking food shortages, as is being witnessed currently. These include non-sustainable irrigation rates, loss of biodiversity, destructive land-use change, and nutrient loss in the soil [4]. The cultivation of energy crops on devalued land sites like closed landfills is a multidimensional approach to fulfil different needs like bioenergy generation to generate revenue for post-closure maintenance of landfills, irrigation with unconventional water resources like treated landfill leachate, and phytoremediation of soil [19–21].

This study aims at contextualizing the idea of growing non-food energy crops on closed landfill sites in particular and marginal lands in general as a measure to decrease the dependence on fertile soil for energy derived from biomass in times of multiple crises. In South Asian countries like India, Pakistan, Sri Lanka, and Bangladesh, management of closed landfill sites is a major issue due to its cost-intensive nature [22]. In this work, the

current situation of landfills in South Asian countries and options for the cultivation of non-food energy crops on closed landfill sites and marginal lands were reviewed. The idea of cultivation of non-food energy crops of marginal lands like landfill sites is an idea that has not yet been widely discussed in the context of South Asia. Hence this paper aims at starting a conversation in this direction, given that the countries in this region are moving towards more sustainable waste management methods that will render many landfill sites needy of post-closure utilization [23,24]. Three energy crops were chosen for this review based on the suitability for cultivation in South Asia, namely, *Ricinus communis* L., *Jatropha curcas* L., and *Populus deltoides*, while as three crops, namely *Miscanthus X giganteus*, *Silphium perfoliatum* L., and *Panicum virgatum* were chosen for the European context in view of their suitability in its different countries (Figure 1). In the European context, for example, Germany has a total of 1005 operational landfills [25] which need to decrease the reduce operations to less than 10% of the total municipal solid waste 2035 according to the Landfill Directive of the European Union [26,27], necessitating a post-use management strategy The comparison of the crops is based on their socioeconomic and ecosystem services provided. The case studies from the European context are also presented to serve as guiding examples for the initiation of such projects in the South Asian region.

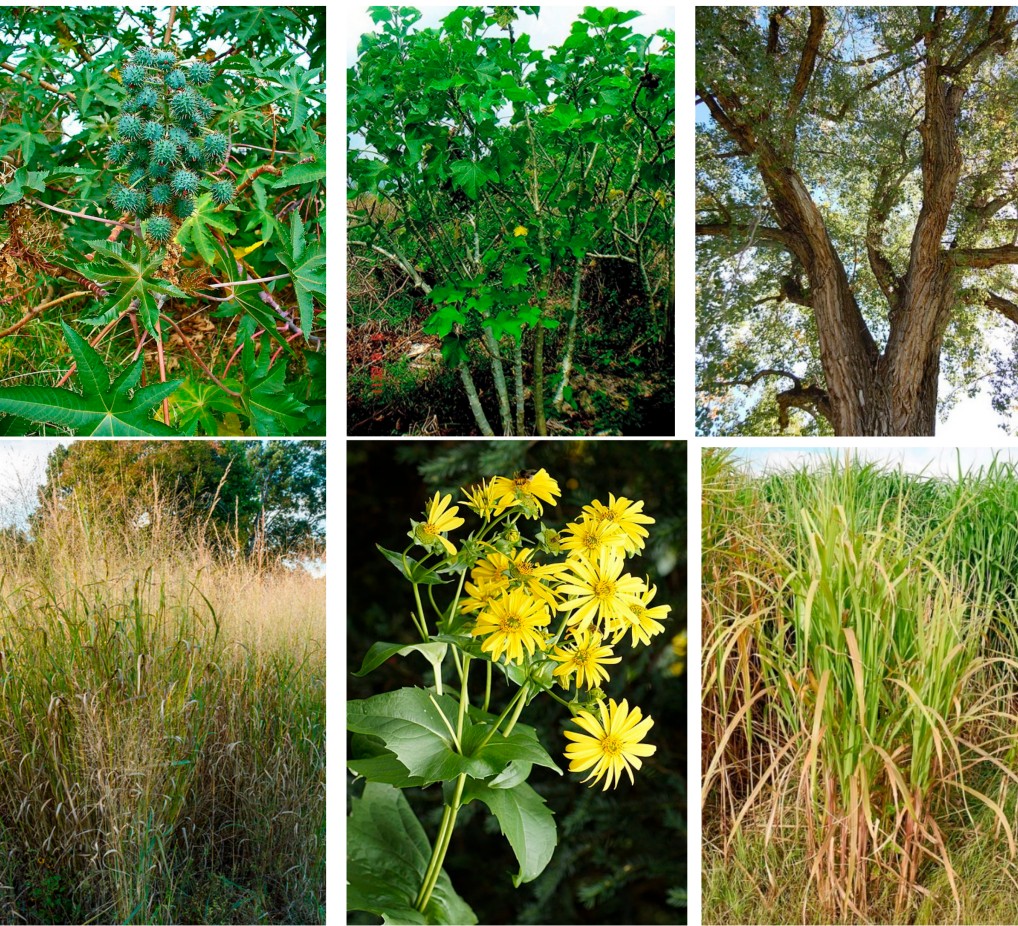

**Figure 1.** The energy crops (clockwise from top left): *Ricinus communis* (H. Zell, CC BY-SA 3.0), *Jatropha curcas* (A. Morad, CC BY-NC-SA 2.0), *Populus deltoides* (M. Lavin, CC BY-SA 2.0), *Miscanthus x giganteus* (K. Schneider, CC-BY-SA-4.0), *Sulphium perfoliatum* L. (U. Schmidt, CC BY-SA 2.0), *Panicum virgatum* (T. Potterfield, CC BY-NC-SA 2.0).

## 2. Landfill Situation in South Asia

South Asia is a population powerhouse, home to one-fourth of the world's population. Landfilling has been the dominant form of waste management in the past few decades and continues to be so. As a result of which many landfills are filled to the capacity,

awaiting closure followed by post-closure maintenance and care. Using closed landfill sites to cultivate non-food crops and trees could possibly open a revenue stream that can contribute to covering the post-closure maintenance and care costs. This is in addition to providing potential ecosystem services like soil phytoremediation, aesthetic enhancements, micro-climate and environment moderation [20]. This strategy is however not restricted to landfill sites but can be equally applied to marginal and degraded lands that are otherwise classified as wastelands [28,29].

This paper proposes a strategy to assist in landfill management in South Asian countries by generating revenue from energy crop plantations on closed landfill sites. For this purpose, the paper discusses the conditions of landfills in South Asian countries in order to compare the current situation and forecast the future potential of possible energy and revenue generation from these landfills.

In order to have a more detailed comparison, the environmental sustainability and cultivation suitability in South Asian countries of the energy crops; *Jatropha*, *Ricinus*, *Populus* discussed in this paper, are reviewed. Finally, the economic, environmental, and functional analysis of the energy crops was done to find out the most optimal option for cultivation on closed landfill sites in South Asian countries. Rapid urbanisation, increasing population and sporadic economic growth in South Asian countries has led to an unprecedented increase in consumption, leading to the major problem of open dumping of a large amount of waste produced. The urban areas in Asia generate around 8 million tonnes of municipal waste per day [22]. The scenario of municipal solid waste (MSW) dumping presents a similar picture in all the South Asian countries, involving unplanned dumping of tonnes of uncovered waste, burning of solid waste, insects and rodent infestation, and waste scavengers picking up valuables from waste [30].

In most of the developing countries, local municipalities do not have enough finances to construct containment landfills, thus problems like absence of top cover, poor site design, lack of leachate collection and treatment systems are common [22]. According to the Global Methane Initiative (GMI), the global anthropogenic methane emissions measure around 8.586 Gt $CO_2$ equivalent and the municipal solid waste division is the fourth major contributor to global discharges of non-$CO_2$ greenhouse gases, especially methane (17%) [31].

In developing countries, due to the lack of enough space for planning new landfills and tremendous pressure of waste disposal in the urban areas, most of the landfills are active even after decades of their operational period [32]. However, in India, many old landfill sites in major cities like Mumbai, Bangalore and Pune have been shut down and converted into recreation grounds, with proper landfill gas extraction system, drainage layer for storm water and vegetation soil layer. Altogether, urban cities in the South Asian countries have the potential to utilize the old landfill sites by the closure and rehabilitation of the landfills which have reached their capacities [32].

## 2.1. Landfill Conditions in Bangladesh

Bangladesh is a country located in the eastern part of the South Asian subcontinent. It is a densely populated country with a population of about 161.4 million. Due to high population density, the agricultural land is saturated with very fewer chances for increment in food production capacity [33]. The increasing population and rapid urbanisation due to economic development has resulted in massive expansion of cities like Dhaka in terms of geographical area and food consumption rate. Factors like urban population growth, economic development, high population density and high consumption rate also contribute to increased municipal solid waste [34]. The disposal of MSW at the landfills is considered as the cheapest means of a solid waste management system. The Matuail landfill is one of the two landfills serving the megacity Dhaka, capital of Bangladesh. It is 27 years old and has reached its capacity [35]. The fact that it is located next to a water body, larger in size than the landfill itself makes the post-closure maintenance and the possible local ecosystem remediation of immense ecological importance (Figure 2) [36].

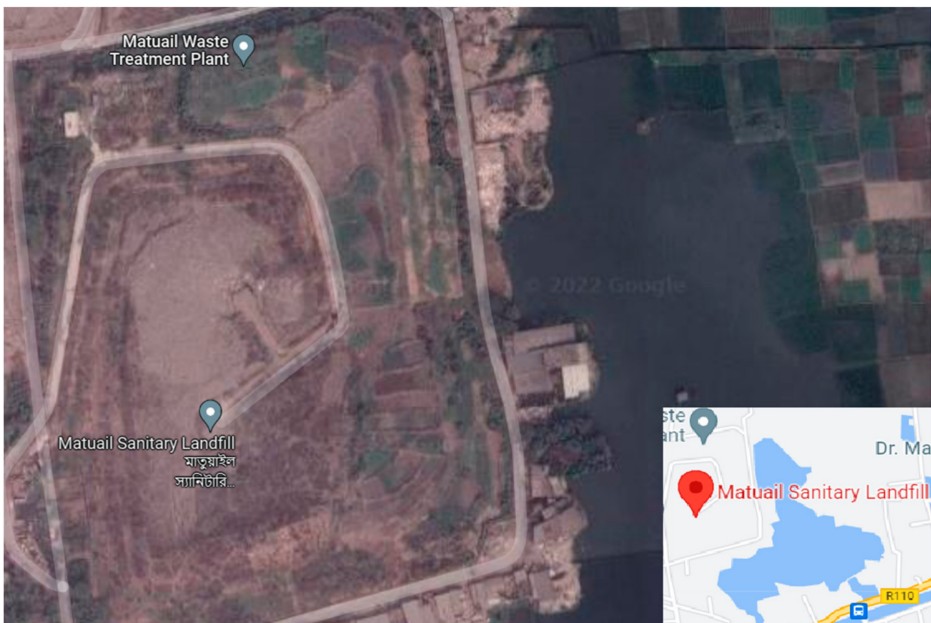

**Figure 2.** The location of the Matuail sanitary landfill in Dhaka, Bangladesh, next to a large water body (Google Maps).

A case study on Matuail sanitary landfill site in Dhaka, which takes about 65% of total waste generated in the city, observed that the treated and untreated leachate samples showed high concentration of COD, $NH_4^+$, $NO_3^-$, $HCO_3^-$, chloride, potassium and certain heavy metals [37]. As a result, continuous monitoring and proper geophysical investigation of the landfill is required for the public health safety [37].

*2.2. Landfill Conditions in India*

India with a population of 1.3 billion, the second-most populous country in the world, is characterised with rapid urbanisation and industrialisation. Unplanned urbanisation coupled with increasing population has led to an increase in waste production in India. As a result of scarcity of land for managed disposal in urban cities, most of the MSW produced is disposed of in overflowing landfills, which poses challenges to public health, environment and land use [38]. India's per capita waste generation varies from 0.2 to 0.6 kg, increasing at a rate of 1.3% per year, which is likely to increase to 5% owing to increase in the urban population [39,40].

According to a study done of 3 landfill sites in Delhi to evaluate chemical and toxicological dangers of leachate in those 3 landfills, it was found that organic component surpassed the upper allowable limit by 158 times [41]. The implementation of landfill gas to energy concept in India involves three main challenges: provision of financial incentives, technology innovation and developing a consistent outline for assessment of the performance. Also, the health impact of old landfills and financial benefits from waste to energy and closing down of old landfills should be a part of government policy [42].

*2.3. Landfill Conditions in Pakistan*

Pakistan has the total area of 796,095 km$^2$ and is one of the most densely populated countries in the world with a current population of 212.2 million. About 36% of the country's population is living in urban areas. As a result of increasing urbanisation, municipal authorities are unable to cope up with the increase in MSW production [31]. About 67,500 tonnes/day are MSW is generated in Pakistan but only 50–69% of it reaches landfills [31]. Mostly the landfills are unlined or non-cemented and open dumping is a common practice [43]. The landfill workers and inhabitants living in areas adjacent to these

landfills are under serious threat due to air, soil and water pollution resulting in fungal infection, diarrhoea, and ulceration of skin [43].

A study was done at Mehmood Booty landfill situated in Lahore to assess the ecological risk associated with the dumping site which receives around 1200–1500 tonnes/day of solid waste [44]. It was found out that leachate of this landfill contained a much higher amount of total dissolved solids, total suspended solids, hardness and alkalinity, as compared to FEPA standards [44]. Respiratory diseases were found to be the major problem among the issues faced by the inhabitants residing in the vicinity of the landfill. High levels of heavy metals such as arsenic (0.05 mg/L), nickel (0.051 ppm) and lead (0.067–0.69 ppm) were detected; the presence of these metals in the high quantity indicated the presence of toxic waste in the landfill [44]. In Pakistan, around 30% of anthropogenic methane emission are from landfills, which also release carbon dioxide, nitrogen, and carbon monoxide in a large number [45].

In a study published in 2018, the Muhammadwala landfill in Faisalabad, Pakistan was investigated to estimate the greenhouse emissions from the landfill over a 28-year time frame by using LandGEM software. The landfill is surrounded by agricultural land (Figure 3) and the open dumping nature of the site poses a direct threat to the agro-ecosystem and the people working around. That this landfill site is now 30 years old and is without a soil cover points to the relevance of closure strategies like the one proposed in this paper. Furthermore, a study reported that MSW disposal sites in Pakistan are responsible for 12.8 million-tonnes $CO_2$-eq. methane emissions annually [46]. According to a study if 75% of the total waste is collected and 50% of this collected waste is landfilled properly, Pakistan has a potential to produce around 83.17 MW of power, which alone could overcome 1.4% current power shortage in the country [31]. The government and concerned authorities should take required measures to ensure proper collection, transportation and disposal of MSW in well managed and designed landfills, and to make the most out of the waste to energy potential from these landfills.

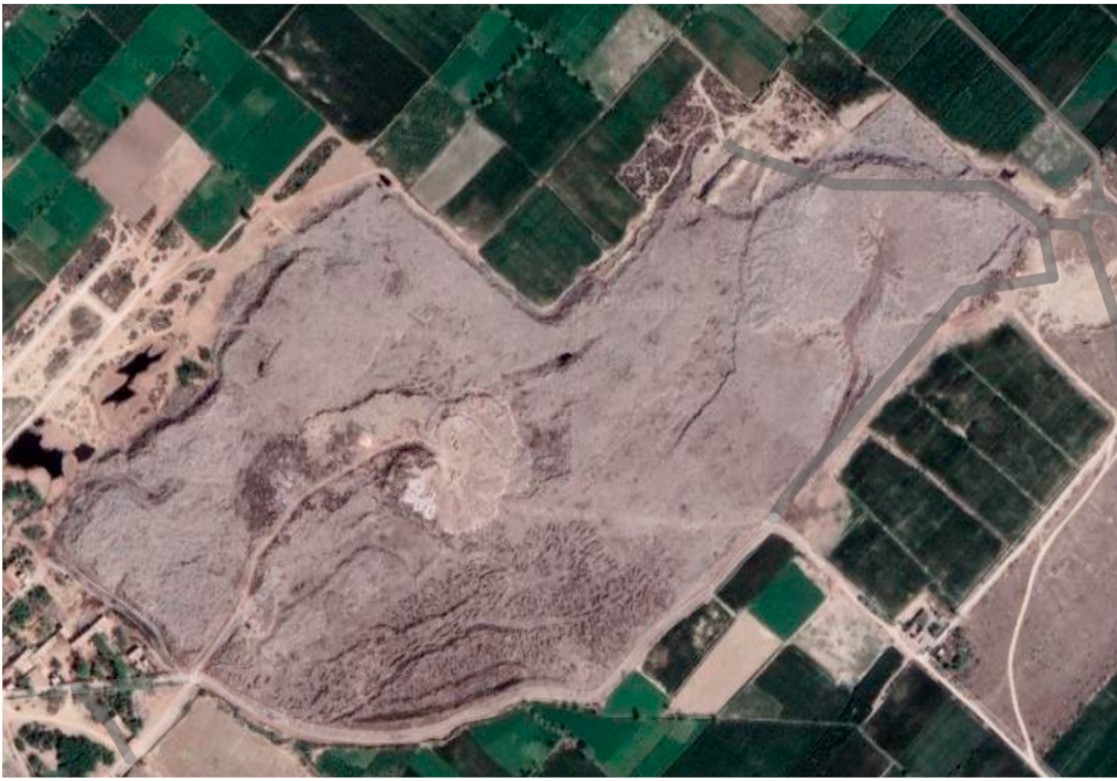

**Figure 3.** The location of the Khurriawala landfill site (Muhammadwala village) in Faisalabad, Pakistan in the midst of agricultural land.

*2.4. Landfill Condition in Sri Lanka*

Sri Lanka, a developing country with a total land area of 65,610 km$^2$ and a population of 21.67 million is an island situated to the south of India. The issues of limited land space and rapidly increasing population density being the limiting factors, the disposal of MSW is a major challenge for the municipal authorities. Inappropriate management of landfill sites in Sri Lanka poses a public health risk and has led to various environmental and social problems [47–50]. According to a study, by the year 2025, the urban municipal waste generation in the country would reach 1 kg/capita/day [49,51].

A study done in Moratuwa, a suburb municipality in Sri Lanka, to investigate current MSW management practices and environmental related problems, reports that waste dumping practices in the municipality caused extreme social and environmental problems. The main problems identified were landfill gas and leachate generation, which are released without treatment directly into the ecosystem [49]. Two different studies in Batticaloa and Gohagoda, Sri Lanka were done to assess the influence of existing landfill sites on groundwater quality due to leachate seepage [52]. The former found out that the concentration of EC, TDS, total hardness, $PO_4^{3-}$, $NO_3^-$, coliform population, BOD, Cu, COD, and Pb were higher than the permissible limit, which makes the groundwater in the vicinity of the landfill unsuitable for drinking [52]. The latter study concluded that leachate from Gohagoda landfill was in methanogenic phase with 7.9 pH, and parameters like $NO_3^-$, Cu, Fe, Ni, CO in the leachate exceeded the allowable limit, also polluting the groundwater nearby [50].

It was found that in Gohagoda dumpsite greenhouse gasses emissions were significant amounting to 2.61 Gg/year, and by using recent waste to energy technologies like gasification, incineration, pyrolysis process around 6.88 GJ/tonne of usable energy could be produced while mitigating the pollution levels [53]. The motivation of reducing emissions and pollution levels from landfills, and generating energy from waste simultaneously could provide directions to adopt a proper landfill management scheme.

## 3. Non-Food Energy Crops on Marginal Lands: The Economics and Ecologics

Energy crops make up a relatively small but important portion among the total biomass produced each year and the amount is expected to grow over the years. cultivation of energy crops however competes with other agricultural activities in terms of resource consumption. As of the current agricultural system, around 70% of water is consumed in agriculture activities, and around 19% of the increase in this figure is expected by 2050 [54]. With a significant proportion of world's vegetable oils and grains being used for the production of biodiesel and bioethanol respectively, in view of the limited land available and recurrent food shortages, an alternative strategy is needed [6,7,9]. In this regard, utilising marginal lands for cultivating non-food energy crops, generating energy from growing non-food energy crops on landfills and phytoremediation of the landfill site simultaneously could be a viable option [19]. The cultivation of crops on contaminated landfill sites can be a solution to bioenergy production while contributing to economic, social and environmental sustainability [4]. Energy crops like *Ricinus*, Switchgrass, and *Miscanthus* thrive even in soils with low fertility and can hence be suitable options for rehabilitated closed landfill sites [55,56]. Owing to food shortages due to multiple crises, different countries are currently exploring potential non-food energy crops for biofuel production and reducing the portion of food crop produce that may be used for energy production. As things stand, energy crops contribute about 1.5% electric power, 3% heat generation and 3% liquid transport fuel throughout the world [57].

*3.1. Populus, Ricinus, and Jatropha Cultivation on Landfill Sites in South Asia*

Some energy crops like *Populus* sp., *Ricinus* and *Jatropha* are being utilised all over the world especially in Southeast Asia, China and India to generate biofuel [57]. Many other annual energy crops like Kenaf, Switchgrass, Canary reed grass, Eucalyptus and Black locust are used for phytoremediation and biodiesel production even in heavy metal contaminated soils [58]. In this study, *Populus*, *Ricinus* and *Jatropha* were selected over other

energy crops because of ease of availability in South Asian countries, the suitability of cultivation and other economic benefits, which are discussed in the latter part of this work. The relevance of bioenergy use over fossil fuel in the context of a developing region like South Asia is that, it causes 10–20 times lower emission than the latter [58].

The production of energy crops has the potential for various environmental, social and economic benefits, as summarised in Figure 4. The multiple potential benefits include restoration of the local soil ecosystem, which can lead to improved biodiversity and aesthetic value of the locality. Furthermore, in addition to being a clean energy source, this can free other fertile lands for food crop cultivation, leading to local economic activity on one hand and improved food security on the other hand. The local community or authorities can also use the cultivation of the crops on the degraded land sites as an opportunity to trade carbon credits leading to increased revenue for the local social economy and better local infrastructure. The contextualisation of the different environmental, social, and economic benefits of this idea can provide the necessary impetus for the initiation of such projects in the South Asian region where such conversations have not yet found ground. While as it can encourage the growth of non-food energy crops on marginal lands like neglected industrial sites as well as closed landfill sites in the European context.

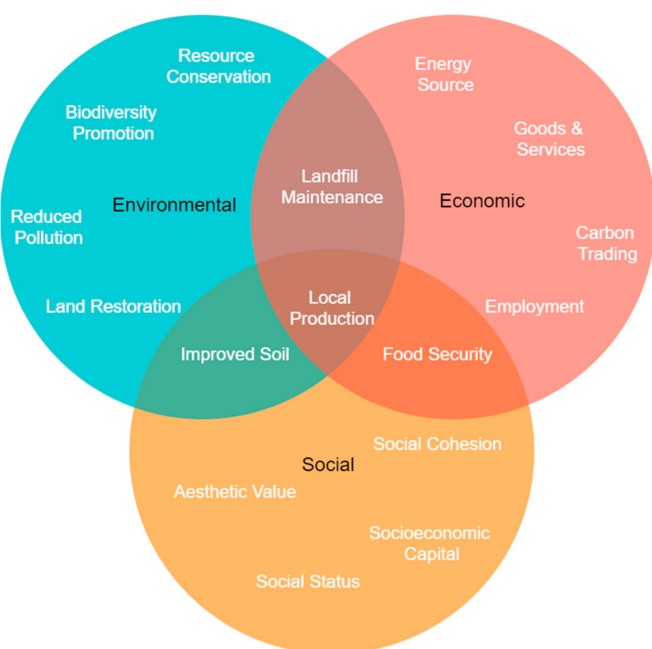

**Figure 4.** A summmary of the different social, environmental, and economic benefits of growing non-food energy crops at closed landfill sites.

The following sections discuss the potential of three perennial energy crops; *Jatropha*, *Populus* and *Ricinus* on the closed landfills owing to their suitability to the South Asian soil type and climate.

### 3.1.1. *Ricinus* Properties, Costs and Benefits

In order to reduce carbon emissions and its impact on the environment, there is a growing concern which has led countries to numerous agreements to develop renewable energy with less impact on the environment. Biodiesel, being biodegradable and non-toxic, is one of the sustainable alternatives with less impact on the environment [59]. Biodiesel is produced from triglycerides by transesterification and fatty acids by esterification, obtained from vegetable oils, algae and from beans like *Ricinus*, *Jatropha*, *Helianthus* and *Pongamia* etc. [60].

*Ricinus communis* L. or castor bean plant is of great interest for the production of biodiesel in South Asian countries, because of tropical climate which facilitates the cultiva-

tion of this crop. India exports about 80% of castor produced in the world, thus making it dominant in the castor oil export market [56]. *Ricinus communis* or castor bean plant has low production cost and is easy to grow, with the capability of yielding 1180 kg oil per hectare and oil contents of 53% [61]. The cost of castor oil as per June 2017, is marketed as US$ 0.66 per kg [61].

According to a case study conducted in the Manabi province of Ecuador, the economic performance of castor oil production was found to concur an average profitability of 47.1% per year [62]. Furthermore, a qualitative assessment to assess the environmental impact of castor biofuel biomass reported major positive impact in contributing to solving different environmental challenges like desertification, soil erosion, invasive species proliferation [62]. A case study field experiment in Tehran, Iran investigated the energy balance and economic feasibility of castor (*Ricinus communis* L.) production [55]. The energy use efficiency in castor seed production was found to be 3.81, which was better as compared to other crops used to produce biodiesel. The study found that the profits from grain yield of 2100 kg/ hectare were upto 3004.03 US$/hectare, depending on the mode of cultivation (Table 1), as depicted in Figure 5. The cost of production per hectare varies from 177 to 454 US$, while as the price per kg of oilseeds in Iran is 1.51 US$ [63]. The biofuel production from *Ricinus communis* is an economical and environmentally friendly option to cultivate on closed landfill sites in South Asia.

**Table 1.** Cost of production, cost per kg of seed and profits from the sale of grains of castor (*Ricinus communis*) grown over a period of two years in Tehran, Iran [42].

| Mode | Production Cost per kg of Seed (US$) | Price per kg of Seed (US$) | Cost of Production (US$/hectare) | Profits (US$/hectare) |
|---|---|---|---|---|
| Using farmer machinery, family labour and subsidised fuel | 0.08 | 1.51 | 177.78 | 3004.03 |
| Using farmer machinery, family labour and unsubsidised fuel | 0.08 | 1.51 | 182.47 | 2999.34 |
| Using Agricultural Service Center services | 0.21 | 1.51 | 454.31 | 2727.5 |

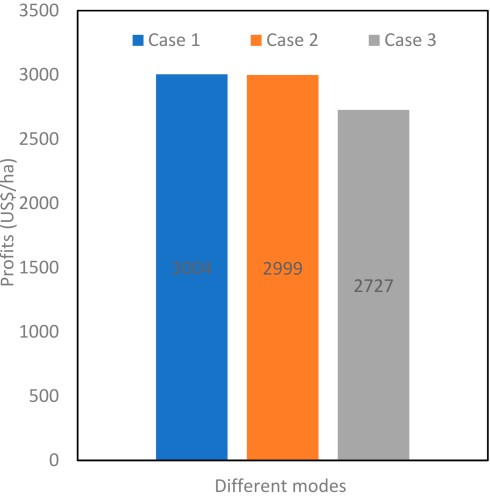

**Figure 5.** A comparison of net profits from *Ricinus* cultivation for three different modes as described in Table 2 [63].

### 3.1.2. *Jatropha* Properties, Cost and Benefits

The present-day need for biofuel, mainly the liquid biofuels like ethanol and biodiesel, has been felt by most of the countries today and they are putting in investments and efforts to promote it. Mostly, edible oilseeds are being used as feedstock by the biodiesel producing countries, for example, rapeseed and sunflower in Europe and soybean in the USA [64].

However, the adoption of the same strategy by developing nations in South Asia can be a difficult task. The reasons would be that the domestic consumption demand would exceed domestic production and risk of shifting cropping pattern from food grain to non-food grains. Thus, a better option for such developing countries would be to cultivate non-edible oilseeds which could be easily grown on degraded or marginal lands [64].

The advantage of using *Jatropha curcas* L. (JCL) as a feedstock is that it can grow in tropical and sub-tropical climates across the developing world, in addition to having a strong adaptability to harsh environments, i.e., high drought resistance, better seed yield and survival rate. The JCL oil is highly viscous and cannot be used directly in engines but only after blending with fossil diesel, or processing into methyl ester; after a two-step process [65,66]. The associated cost with the *Jatropha* cultivation includes establishment cost, operation and maintenance cost and other associated costs like harvesting, seed preparation, transportation and marketing. The cost of biodiesel production from *Jatropha* (average 0.46 US$/litre) is less than the associated costs of biodiesel from oilseeds in the USA (0.50 US$/litre) and EU (0.62 US$/litre), which is another motivation for cultivating *Jatropha* for bioenergy [47,65]. The cost of biodiesel production from *Jatropha* can further sink to 0.40 US$ if by-products like glycerine and animal seed cake can be marketed [47].

A study, in which economic benefits of *Jatropha* plantation were calculated in North-East India, concluded that *Jatropha* plantation, although not being highly profitable, is still economically viable [67]. The overall income per hectare of JCL plantation is tabulated in Table 2 and depicted in Figure 6. The future research and development efforts on seed varieties, the better market price in South Asian markets would make *Jatropha* plantation more attractive [67].

**Table 2.** Costs and returns from one hectare of *Jatropha* plantation under low (L) and high (H) yield scenarios over 10 and 40 year periods [45].

| Costs and Returns | 10 Years | | 40 Years | |
|---|---|---|---|---|
| | Scenario L (US$/hectare) | Scenario H (US$/hectare) | Scenario L (US$/hectare) | Scenario H (US$/hectare) |
| Total cost on JCL plantation | 560 | 1430 | 3150 | 8050 |
| Gross Income | 1050 | 3230 | 6100 | 18,050 |
| Net Income | 490 | 1800 | 2950 | 10,000 |
| Benefit-Cost ratio | 1.25 | 1.88 | 1.64 | 2.13 |

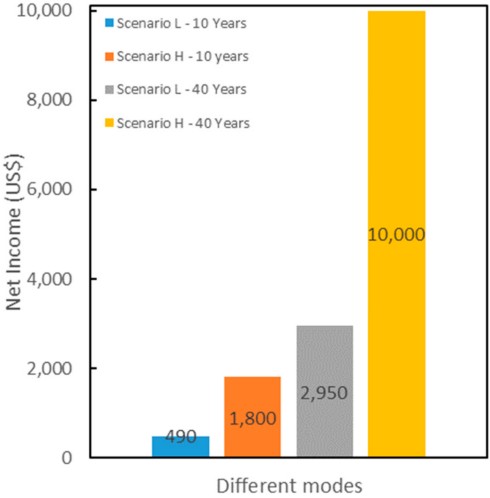

**Figure 6.** A comparison of net income from *Jatropha* cultivation in different scenarios (L: Low yielding scenario; H: High yielding scenario) [67].

### 3.1.3. *Populus* Properties, Cost and Benefits

In order to develop sustainable and local energy sources, biomass has received attention because of its capability to reduce dependence on external sources and carbon emissions [68]. Dedicated energy crops like *Populus deltoides* are gaining popularity internationally to produce biomass as compared to other extensive energy crops because of the high yield and ecological benefits in terms of low input necessities and biodiversity maintenance [68]. A study was done in the eastern and northern part of India, which determined the energy content of branches, stems, roots and litter of *Populus* on plantation over two ages, i.e., 5-year-old (3924.15 hectares) and 7-year-old (2386.37 hectares) [69]. In this study, using a bomb calorimeter, it was found that the energy stored in the above-ground *Populus* tree components from 2131.87 ha (5-year old) or 1002.88 ha (7-year old) was enough to operate a 5 MW generating station for a year. Furthermore, it is reported that the energy stored (898.55 GJ) in the above-ground biomass from 1 ha of 7-year-old poplar plantation is enough to meet the electrical energy needs of an average household in eastern India for 18 years [69].

Another study was done to assess the feasibility of implementation of large-scale cultivation of *Populus* in Spain [68]. An interesting finding of this study was regarding the water consumption angle to bio-energy production. It was reported that the water consumption per unit of energy obtained was $45 \, \text{m}^3 \, \text{GJ}^{-1}$ while as the water required to avoid a kg of $CO_2$ emissions is $4.6 \, \text{m}^3$ [68]. The energy balance and energy efficiency obtained from this study are depicted in Table 3. It was found that the low-density system was the most efficient option among the three options [68]. Assuming the current consumption of firewood to produce fuel by a medium-sized family in South Asian countries to be 7.5 kg/day, with an average energy value of 18.5 kJ/g or 4.16 GJ/month, the equivalent energy utilised is 203 KWh/month. The monthly cost for energy consumption was assumed to be around 16.24 US$ per month (1 KWh = 0.08 US$) [68].

**Table 3.** Energy balance of the *Populus* bioenergy system for two years [68].

| | Low-Density System (GJ ha$^{-1}$) | High-Density System (GJ ha$^{-1}$) | Gas Natural System (GJ ha$^{-1}$) |
|---|---|---|---|
| Energy input (A) | 31.1 | 46.7 | 53.5 |
| Energy output (B) | 292.1 | 351.1 | 292.1 |
| Energy balance (B-A) | 261.0 | 304.4 | 238.7 |
| Efficiency (B/A) | 9.4 | 7.5 | 5.5 |

The profits generated from a 7-year-old plantation of *Populus* over 2386.37 hectares of land, is depicted in Tables 3 and 4. However, economic feasibility and plantation operations needs to be further studied extensively before implementation.

**Table 4.** Expected profits from *Populus* plantation based on the above ground tree biomass [69].

| Area of the Plantation (7-Year-Old Plantation) | 1002.88 Hectares |
|---|---|
| Energy Produced (For a 5 MW power plant) | 901,141.2 GJ/year |
| Cost per Giga Joule of energy (approximate) | 25 US$ |
| Total earnings (US$) | 22,460 US$/hectare |

### 3.2. Energy Crops on Marginal Lands in European Countries

On the European level, 17.5% of the gross energy consumption is accounted for by bioenergy and more than half of the renewable energy consumption can be traced back to bioenergy sources [70,71]. Maize and *Miscanthus* are the main bioenergy crops used in many countries in the European region [71,72]. In view of the challenges posed by the growth of food energy crops like maize with respect to food shortages, the use of non-food energy crops that have less soil fertility needs is being explored as an option. In Europe, a substantial land area is already classified as marginal while as more area

is becoming increasingly marginal due to factors like erosion, salinity, drought, land use change, and pollution [73]. To avoid competition for productive land and healthy soils between biomass production for energy and food/feed production for humans and animals, it is recommended to focus on marginal lands or contaminated soils to grow energy crops [20,74,75]. This is pertinent given the land mass under biofuel cultivation in Europe is around 14 million hectares, an area larger than the total land area of Greece [6]. From the consumption point of view, 58% of all rapeseed oil, 50% of palm oil, and 32% of soy oil ends up being used as biofuel in automobiles [7]. In order that the crops are then exclusively used for energy generation and not as food, the selected crops have to be of non-food nature. Furthermore, such cultivation can also be utilized for the phytoremediation of contaminated soils [75–77]. Lignocellulosic crops like *Miscanthus* and switchgrass exhibit a lower baseline yield gap between cultivation in healthy soils and marginal soils; another argument for using marginal soils for cultivating such energy crops. In a long-term field trial conducted in England, energy crops like *Miscanthus*, reed canarygrass, and switchgrass were cultivated on different marginal land sites including a closed landfill site, and bioenergy generation up to 97 GJ per hectare per year was achieved [78]. In the following sub-sections three non-food bioenergy crops are discussed in the European context.

### 3.2.1. *Miscanthus X Giganteus*

*Miscanthus* is a perennial lignocellulosic crop with a potential to increase profits from bioenergy feedstock production in European rural areas [73]. It has low input requirements and high above and below ground biomass productivity [74,75,79,80]. Furthermore, it is characterised by high stress tolerance as well as high phytoremediation potential [75,81–83] for heavy metals like Cadmium, Mercury, Lead, Arsenic, Chromium, Copper, Nickel, and Zinc [75,84–87]. The occurrence of more accumulation in root zone as compared to shoot zone is beneficial to design different biomass processing processes for different usage streams [83]. Bilandzija et al. [75] did a study with *Miscanthus* on soils contaminated with heavy metals mercury and cadmium and concluded that they did not significantly affect the combustion properties of *Miscanthus* [75]. Furthermore the analysed parameters with the exception of Cd, K, and Cr were found to be within the ISO limits for soild biofuels or with expected minor deviances from data found in literature [75]. *Miscanthus* is also considered beneficial in mitigating soil erosion and is a viable method of achieving high amounts of carbon sequestration [73,88].

Certain species of Miscanthus have been found to be capable of growing in the harsh Siberian climate [89,90] however the yield has been seen to vary significantly between different climates [91–93]. In temperate climate, three-year old *Miscanthus* plants demonstrated a bioethanol yield of up to 5600 L/hectare which translates to 120–200 L/t, given a per hectare yield of 30–50 tonnes [81]. United Kingdom leads the list of countries with significant *Miscanthus* cultivation with 10,000 hectares followed by France and Germany with 4000 hectares each [89,94]. The total marginal land area in Europe suitable for the cultivation of *Miscanthus* is estimated to be 11.11 million hectares, which indicates that the potential for growth in this sector has yet to be tapped [89]. Studies however have pointed to the tedious method of crop established as the main cost factor and impediment for the widespread adoption of *Miscanthus* despite its evident profit potential [89,95].

### 3.2.2. *Silphium perfoliatum* L.

In addition to the commonly used energy crops of *Miscanthus* (for solid biofuel) and maize (for biogas), there has been a growing interest in perennial herbaceous crops like *Silphium perfoliatum*, which bypass the challenges associated with the growth of the commonly used energy crops [70,96]. *Silphium* possesses distinctive characteristics that make it an ideal energy crop in this context. These include the possibility of harvesting over many years without annual planting, ease of harvesting with common farm equipment, and a boost to the on-farm biodiversity [70]. This plant produces annual yields upto 15 years after establishment starting from the second year of planting [70,97]. Initially

explored in eastern European countries [70], this plant has now caught interest all over the European Union [96,98]. In year 2020, it was grown on over 400 hectares of land in Germany alone [70,98]. Owing to its environmental benefits, this plant has been listed as an eligible species in the Ecological Focus Areas by the European Commission [70,99].

The annual yield of *Silphium* has been reported variously in the range of 12 to 23 tonnes dry mass per hectare [96]. The average yield at high altitudes has been reported as 15 tonnes dry mass per hectare, which is expected to increase with increased water availability, prompting the observation that *Silphium* 'can compete with current energy crops in terms of dry matter yield' [70,98,100]. The average calorific value for *Silphium* ranges from 17.7 MJ/kg for stems and 17.4 MJ/kg for pellets [70]. In Germany, it has mostly been used as a biomass stock for methane production through anaerobic digestion as well as an alternative to maize as a biogas feedstock [101,102]. The methane yields of *Silphium* differ from those of maize by only 5–10% and it is hence seen as a viable alternative from both economic as well as ecological point of view [102,103]. Furthermore, cultivation of *Silphium* has been reported to improve soil carbon sequestration [96] The idea of growing *Silphium* together with maize as a cover crop has already been patented in this regard [70,102].

### 3.2.3. *Panicum virgatum* (Switchgrass)

Switchgrass is a perennial grass which can provide cellulosic feedstock for bioenergy production [104]. It is a non-food energy crop with a high carbon sequestration potential and can lead to improvement in soil quality in addition to being a clean energy source [88,104]. It's cultivation on marginal lands can also lead to a reduction in non-point source pollution that would otherwise result from conventional crop land [88,105–107]. It has the potential to be cultivated across the different soil types in Europe and adjust well to marginal soils and drought conditions majorly due to its extensive root system [88,105,107]. Due to its low water and nutrient requirements and high potential yields even in marginal soils, it is an ideal candidate for a cross-regional planning of its use as a non-food energy crop in Europe [108]. The bioethanol yield for switchgrass has been reported to be of the order of 100–249 L/t which corresponded to a per Liter cost of less than US$ 1.1 which is a competitive price compared to crude oil price [108]. Furthemore, regarding comparative greenhouse gas emissions assessment, it has been reported that switchgrass shows a net reduction compared to gasoline and has been mentioned as an effective substitution for fossil fuels [108].

Furthermore, switchgrass has been reported to grow well in soil contaminated by heavy metals with less or no negative effect of metal contamination [109]. This may make it a favourable bioenergy crop candidate for soils in and around closed down industrial sites. A study trial conducted by Xiufen Li et al. [104] concluded that switchgrass cultivation mantains or increases the soil nitrogen stocks [104]. Walter Zegada-Lizarazu et al. [17] also reported an increase in the nitrogen stock of marginal soil after switchgrass cultivation while also reporting a marked increase in the soil organic carbon (SOC) stocks in soil [17]. These studies point to the suitability and the benefits of cultivating switchgrass as a dedicated energy crop in marginal soil.

## 4. Discussion

The cultivation of non-food energy crops on marginal land sites like closed landfills, is an idea aimed at generating revenue from the biomass and biofuels produced from energy crop plantation [28,29,65]. At the same time, this course can contribute to phytoremediation of the land, aesthetic development, and microclimate moderation, in addition to covering some costs of post-closure maintenance and aftercare. In this study, three non-food energy crops; *Ricinus*, *Jatropha*, and *Populus* were explored based on possible profits that could be generated from their cultivation on closed landfill sites and their environmental benefits, based on existing literature. This was discussed in the context of the landfill situation in South Asian countries. While as *Miscanthus X Giganteus*, *Silphium perfoliatum* L., and *Panicum virgatum* were reviewed in the context of Europe.

### 4.1. Economic Benefits

As previously mentioned in this paper, the estimate of 200,000 US$/hectare for a 30 year-long post-closure maintenance and care has been widely reported [110]. This translates to yearly maintenance costs of around 6666 US$. The economic returns of the three discussed energy crops are a viable means to cover these costs in the long term, in part or whole, depending on the different modes of cultivation employed. With *Ricinus* cultivation having a potential of providing 3000 US$ per hectare with a two-year-old crop, it appears to be the fastest option [55]. The other two crops discussed are long term investments; *Jatropha* provides 1800 US$ per hectare over 10 years and 10,000 US$ per hectare over 40 years, while as a *Populus* plantation promises 22,460 US$ per hectare over the period of 7 years [69]. Given that the mandatory post-closure maintenance period is of 30 years, a long-term investment like the cultivation of energy crops promises to be a dependable source of revenue (Figure 7) [55,63,65,67,69]. The apparent long waiting time for profit generation in case of *Jatropha* and *Populus* in particular, hence, is not of great disadvantage, when the discussion is about tracts of land that are devoid of any potential to grow any other (food) crops and might otherwise be classified as waste lands.

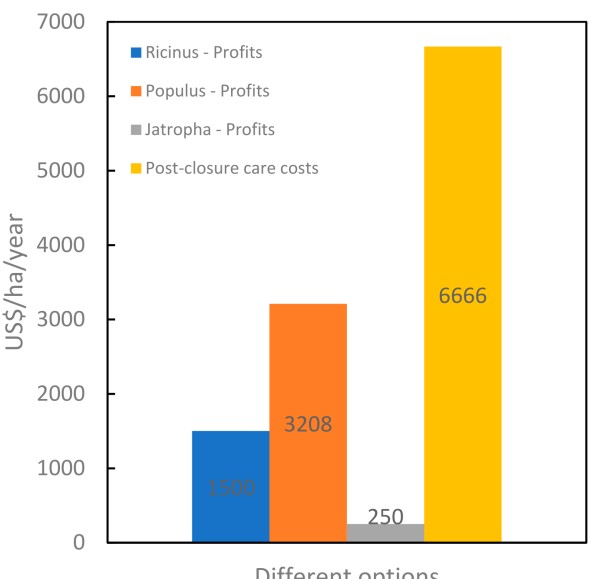

**Figure 7.** A summary of the reported profits from *Ricinus*, *Jatropha* and *Populus* cultivation and the post-closure maintenance costs in US$ per hectare per year [55,63,67,69].

Considering a conservative price of US$ 66 per tonne dry matter biomass, from a purely economic point of view, *Miscanthus* is the frontrunner with a per hectare yield average of 40 tonnes [81]. However, Switchgrass has been reported to have a better conversion factor with around 200 litre bioethanol per tonne compared to 160 litre for *Miscanthus* [81,108]. *Silphium* however offers a competitive alternative with yields of up to 25 tonnes per hectare and the possibility of being used as an intercrop with other established bioenergy crops like maize [70,102]. A comparative study of different bioenergy plants in Europe reported that *Silphium* was more profitable than *Miscanthus* in trials in UK, Germany, and Poland [96]. Furthermore, the ecological benefits and the long-term yield generation potential of Silphium makes it a more viable candidate as is evident from the growing interest in the plant in the European market. However, it is pertinent to mention here that water stress susceptibility is a factor that may favour the other two crops as compared to *Silphium* [96,111].

From an economic point of view, it was found that *Populus* is a better long-term option for generating revenue from the biofuel produced in the South Asian conditions [68]. For a short-term strategy, *Ricinus* provides quicker returns than the other two crops [60,62].

However, the high survival rates in adverse conditions for *Jatropha* tilt the balance in its favour depending on the type of land under consideration [65]. From an environment and remediation point of view, all the three discussed energy crops are highly beneficial in South Asian countries. *Ricinus* crop has high medicinal value and high stress tolerance when compared to other energy crops discussed in the paper [60]. However, considering the high biomass production, phyto-stabilising, and phytoremediation potential of *Populus* on heavily contaminated land sites like landfills, the cultivation of *Populus* could be a better long-term option [68]. From the case studies, it can be observed that the revenue-generating potential of these crops can be optimised with different crop establishment strategies. Hence, agronomic considerations, in addition to socioeconomic and environmental considerations can play a role in the selection of the crops or plants (single or multiple) better suited for a particular site.

The lower fertiliser and pesticide requirements of *Miscanthus* make it an economically viable bioenergy crop [81,95]. However, its high establishment costs and problems associated with processing like corrosion weaken its candidature as the bioenergy crop of choice. The higher abiotic stress tolerance improves the suitability of *Miscanthus* for cultivation in marginal areas [81,95]. The establishment costs of *Miscanthus* however are expected to decrease with the dawn of direct seeding crop establishment method. This is expected to lead to its wider adoption among smaller budget farmers. In regions with lower expected abiotic stresses, *Silphium* is a better alternative to *Miscanthus* and can even compete with maize in terms of biomass yield and environmental benefits like biodiversity improvement and carbon sequestration [70,80,96,111]. The low establishment costs of *Silphium* and its ability to be grown as a cover crop together with maize can prove to be the gamechanger in the use of nonfood crops for bioenergy generation. The low drought tolerance of *Silphium* however is a factor that needs to be considered when selecting *Silphium* as the bioenergy crop of choice.

### 4.2. Ecosystem Services

In addition to the revenue generation incentive, cultivation of non-food energy crops also provides other benefits. Environmental and ecosystem benefits including phytoremediation, carbon sequestration, microclimate moderation, aesthetic development, and biomass generation have the potential to make the idea more acceptable in South Asian countries. A comparative summary of these different benefits of RCL, *Populus*, and *Jatropha* is presented in Table 5.

**Table 5.** Environmental benefits and suitability to South Asian climate [35,41,52,53].

| *Ricinus communis* L. | *Jatropha curcas* L. | *Populus deltoides* |
|---|---|---|
| High physical, biological and climatic stress tolerance | Drought and pest tolerant | High adaptability in several sub-tropical and tropical regions |
| Good growth on abandoned, waste and barren land like landfills | Good growth in degraded lands; requires less maintenance | Growth capability in heavy metal contaminated soil; high biomass. |
| A native of tropical Asia and Africa, highly suitable for South Asian countries | Native to Mexico and Central America, but cultivation present through Africa, South-East Asia | Native to North America, Europe and Asia, grows from tropical to temperate regions |
| High medicinal value | Used for traditional medicinal practices | The bark, leaf and leaf buds medicinally valuable |
| Contains 90% ricinoleate, high-profit source of hydroxyl fatty acid | High seed oil content (30–35%) | High biomass production, high heating values |
| Capability to enrich the soil and prevent soil erosion | Highly effective in the prevention of soil erosion | Phytoremediation of soil containing Cd, Cr, Cu, Pb |
| Highly effective in phytoremediation of soil | High carbon sequestration. Excellent phytoremediation against heavy metals and pesticide contamination | High potential against contaminated wastewater, leachate landfill and waste flows. |

In the European context, *Silphium* is presented as an ecologically sounder alternative to the other established energy crops [70,96] potentially ameliorating environmental costs associated with the established bioenergy crops e.g. biodiversity loss in case of maize [70]. Long term *Silphium* cultivation has been reported to have the potential to contribute to local biodiversity enhancement, improving soil quality, controlling erosion, and improving water infiltration [111]. From an establishment costs point of view, *Silphium* provides the option of being established by sowing rather than by transplantation of seedlings, which can considerably save establishment costs [111]. In this regard, in some cases, the net revenue of *Miscanthus* was found to be negative owing to its high establishment costs [96].

This study is limited in its scope as it doesn't consider the specific country-wise conditions of the different countries in these two regions. It is rather a general assessment of these two regions and an attempt to contextualize the idea of using marginal land areas like landfills to cultivate non-food energy crops in the current scenario in which multiple crises have disrupted the food and energy supplies and raised the prices of food, fuel, and fertilisers [112].

## 5. Conclusions

This study reviewed the strategy of cultivating non-food energy crops on closed landfills in South Asian countries and marginal lands in European countries as a means for socioeconomically sustainable bioenergy production and to contribute to the landfill post-closure maintenance costs. In the context of South Asia, three non-food energy crops *Ricinus Communis* L., *Jatropha curcas* L., and *Populus deltoides* were reviewed on the basis of successful case studies. For the European context, three plants *Miscanthus X Giganteus*, *Silphium perfoliatum* L., and *Panicum virgatum* were reviewed. The environmental and economic benefits of these crops, in addition to the linked ecosystem services were discussed.

Although the main aim of this study was to suggest a viable pathway to rehabilitate closed landfill sites, the strategy proposed can equally be implemented in other degraded land areas like closed down industrial zones or agricultural land no longer suitable to grow food crops. The non-food energy crops presented in this study are representative of a wider range of options that can be considered for bioenergy production based on the site-specific needs and considerations of the local flora and fauna. In the context of the dramatic increase in the prices of food, fuel, and fertilizers worldwide, the idea of growing non-food energy crops on degraded and marginal land sites while sparing fertile land for food crops has become ever more relevant. While as in case of the European region such case studies have been undertaken this idea has not yet found ground in the South Asian region where conversations are still dominated by the topic of using food crops and fertile land for bioenergy production.

It can be generally concluded that dedicated non-food energy crops cultivated on marginal lands provide a competitive alternative to both the conventional fossil-based fuel options and food crop-based bioenergy production. However, site-specific conditions, local weather and biodiversity, and regional regulations need to be taken into consideration when choosing the right crop. In case of landfills, the size and hence maintenance costs should be taken into consideration before deciding on a short-term or a long-term option. This study could provide pointers towards a strategy for the countries that are facing a major problem of contaminated lands and increasing bioenergy demands by focusing on the soil restoration, phytoremediation, and bioenergy generation by utilizing marginal lands like closed landfill sites for growing low-input and low-risk non-food crops, plants, and trees. Furthermore, in view of the existing research gaps around the use of degraded and marginal lands for bioenergy production in South Asian context, the initiation of such research projects is of increasing relevance in the current times of multiple crises.

**Author Contributions:** Conceptualization, T.M.S. and R.O.; methodology, T.M.S.; formal analysis, T.M.S., C.N. and A.H.K.; investigation, C.N. and A.H.K.; resources, R.O.; data curation, T.M.S. and I.S.; writing—original draft preparation, T.M.S. and A.H.K.; writing—review and editing, T.M.S. and I.S.; visualization, T.M.S. and A.H.K.; supervision, T.M.S.; project administration, R.O. All authors have read and agreed to the published version of the manuscript.

**Funding:** This research received no external funding.

**Informed Consent Statement:** Not applicable.

**Data Availability Statement:** No new data were created.

**Acknowledgments:** Publishing fees supported by Funding Programme Open Access Publishing of Hamburg University of Technology (TUHH).

**Conflicts of Interest:** The authors declare no conflict of interest.

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
