# Peer review of "Using Landfill Sites and Marginal Lands for Socio-Economically Sustainable Biomass Production through Cultivation of Non-Food Energy Crops: An Analysis Focused on South Asia and Europe"

_sustainability, doi:10.3390/su15064923_

Round 1

Reviewer 1 Report

1.      As your abstract's final sentence, include a "take-home" message.

2.      Rearrange keywords alphabetically.

3.      It is unclear whether the author's something new in this review work. According to evaluation, several published studies by other researchers in the past adequately explain the issues you made in the present review paper. Please be careful to highlight in the introduction section anything really innovative in this work.

4.      Previous review has to be explained in the introduction section, including their work, novelty, and limits, to illustrate the review gaps that will be filled in the current study.

5.      In the last paragraph of the introduction, please explain the objective of the present article.

6.      Related to biomass utilization, I am encouraged to you to discuss related to green energy and recycle battry potential. Please refer the relevant reverence as follow: Power and Energy Optimization of Carbon Based Lithium-Ion Battery from Water Spinach (Ipomoea Aquatica). J. Ecol. Eng. 2023, 24, 213–23. https://doi.org/10.12911/22998993/158564

7.      Discussion presented in this article is poor. Major improvement is mandatory especially in discussion the results, not just simply mention the results with brief explanation.

8.      The limitation of the current review must be included at the end of the discussion section.

9.      The conclusion section needs to explain further research.

10.   Five years back literature should be enriched into the reference. MDPI reference is strongly recommended.

11.   The manuscript needs to be proofread by the authors since it has grammatical and language issues.

12.   Graphical abstract is encouraged to provide in submission after review.

Author Response

Dear Reviewer,

I thank you on behalf of all the authors for your comments and suggestions. Please find the responses to your review comments as follows:

  1. As your abstract's final sentence, include a "take-home" message.

Done.

  1. Rearrange keywords alphabetically.

Done.

  1. It is unclear whether the author's something new in this review work. According to evaluation, several published studies by other researchers in the past adequately explain the issues you made in the present review paper. Please be careful to highlight in the introduction section anything really innovative in this work.

This study contextualizes this idea in view of the recent multiple crises and highlights the increased relevance of the same in tackling the crises. Furthermore, in the context of South Asia, cultivation of non food energy crops is an idea that has not been talked about at length. Changes have been made in the abstract and the introduction highlighting this aspect.

  1. Previous review has to be explained in the introduction section, including their work, novelty, and limits, to illustrate the review gaps that will be filled in the current study.

The introduction section and abstract have been accordingly modified.

  1. In the last paragraph of the introduction, please explain the objective of the present article.

Done.

  1. Related to biomass utilization, I am encouraged to you to discuss related to green energy and recycle battry potential. Please refer the relevant reverence as follow: Power and Energy Optimization of Carbon Based Lithium-Ion Battery from Water Spinach (Ipomoea Aquatica). J. Ecol. Eng. 2023, 24, 213–23. https://doi.org/10.12911/22998993/158564

Thank you. I found the phytoremediation (the ability to absorb heavy metals from the growth media, such as mercury, arsenic, zinc, copper, and nickel) aspect of this paper relevant to our manuscript and added its reference.

  1. Discussion presented in this article is poor. Major improvement is mandatory especially in discussion the results, not just simply mention the results with brief explanation.

A restructuring of the discussion and conclusion section has been done in this regard, in addition to some additions.

  1. The limitation of the current review must be included at the end of the discussion section.

This has been included.

  1. The conclusion section needs to explain further research.

Done.

  1. Five years back literature should be enriched into the reference. MDPI reference is strongly recommended.

Thank you for the suggestion. MDPI references are included at appropriate places.

  1. The manuscript needs to be proofread by the authors since it has grammatical and language issues.

An extensive proofreading and necessary restructuring has been done throughout the manuscript. Thank you for your detailed review.

  1. Graphical abstract is encouraged to provide in submission after review.

Thank you for the suggestion. This can be submitted after the review is complete.

Reviewer 2 Report

 This paper provided a review of the economic as well as 29 environmental benefits of growing Ricinus communis L., Jatropha curcas L., and Populus deltoides as 30 energy crops on closed landfill sites in the South Asian context.  Please see specific comments below:

1.     In the paper,the references should be more compact and clear, and pay more attention to the logical relationship between paragraphs.

Eg: Novel economy and carbon emissions prediction model of different countries or regions in the world for energy optimization using improved residual neural network. Science of The Total Environment

2.     Novel risk prediction model for food safety based on random forest integrating virtual sample. Engineering Applications of Artificial Intelligence

3.      Many methods about socio-economically sustainable biomass production should be explained more clearly.

4.     The flaw chart or the process of the paper should be written in detail.

5.     Many comparison analysis should be added.

Author Response

Dear Reviewer,

I thank you on behalf of all the authors for your comments and suggestions. Please find the responses to your review comments as follows:

  1. In the paper,the references should be more compact and clear, and pay more attention to the logical relationship between paragraphs. 

Eg: Novel economy and carbon emissions prediction model of different countries or regions in the world for energy optimization using improved residual neural network. Science of The Total Environment

Thank you. The paper has been extensively proofread and the references have been thoroughly checked and updated wherever necessary.

  1. Novel risk prediction model for food safety based on random forest integrating virtual sample. Engineering Applications of Artificial Intelligence

The mentioned publication is not relevant to this paper.

  1. Many methods about socio-economically sustainable biomass production should be explained more clearly.

The scope of this paper covers biomass production through the cultivation of energy crops on marginal lands like landfill sites. Other methods are not in the scope of this paper.

  1. The flaw chart or the process of the paper should be written in detail.

Changes have been made in the abstract, introduction, discussions, and conclusions accordingly.

  1. Many comparison analysis should be added.

In case of Europe, one more large scale case study has been added to the paper. However, in case of South Asia such comparisons do not yet exist regarding the cultivation of energy crops on landfill sites .

Reviewer 3 Report

Dear Author,

The following are recommendations for your consideration:

1.      The introduction section is less focused, there are too many elements that the author touched such as the water-food-energy nexus, agriculture water consumption, and food insecurity. It is suggested that the author rearranged or summarized the introduction section, so the reader can understand the focus of the study which is the usage of the closed landfill to cultivate non-food crops.

2.      It is good that the author includes the landfill condition sections in section 2. However, the connection or interrelation of the information given in this section with the topic cannot be seen. Maybe the author could add some information on how the species discussed in this study could help to improve the environmental performance that has been mentioned in section 2.

3.      It is not clear why section 2.2 was included in this manuscript. It is suggested the author explain more on the difference between the maintenance cost after landfill closure vs cultivating cost on the closed landfill.

4.      It is suggested the author explain specifically the information summarized in Figure 4 at least in one paragraph because the author mentions multiple times in the text that the production of energy crops has various environmental, social, and economic benefits. For example, explain how food security could be benefitted by the production of energy crops.

Author Response

Dear Reviewer,

I thank you on behalf of all the authors for your review comments and suggestions. Please find the responses to the review comments as follows:

  1. The introduction section is less focused, there are too many elements that the author touched such as the water-food-energy nexus, agriculture water consumption, and food insecurity. It is suggested that the author rearranged or summarized the introduction section, so the reader can understand the focus of the study which is the usage of the closed landfill to cultivate non-food crops.

Thank you for the remark. Changes, additions, and modifications to the text have been made in the abstract, introduction, discussions, and conclusions accordingly.

  1. It is good that the author includes the landfill condition sections in section 2. However, the connection or interrelation of the information given in this section with the topic cannot be seen. Maybe the author could add some information on how the species discussed in this study could help to improve the environmental performance that has been mentioned in section 2.

Changes have been made accordingly to the introduction. The idea of cultivation of non-food energy crops of marginal lands like landfill sites is an idea that has not yet been widely discussed in the context of South Asia. Hence this paper aims at starting a conversation in this direction, given that the countries in this region are moving towards more sustainable waste management methods that will render many landfill sites needy of post-closure utilization.

  1. It is not clear why section 2.2 was included in this manuscript. It is suggested the author explain more on the difference between the maintenance cost after landfill closure vs cultivating cost on the closed landfill.

Thank you for the comment. The section 2.2 has been excluded as it is not necessary in the scope of this paper.

  1. It is suggested the author explain specifically the information summarized in Figure 4 at least in one paragraph because the author mentions multiple times in the text that the production of energy crops has various environmental, social, and economic benefits. For example, explain how food security could be benefitted by the production of energy crops.

Thank you for this suggestion. A paragraph has been added to explain Figure 4.

Round 2

Reviewer 1 Report

Well done and I think it is enough in the present form.